# Chemical Constituents of *Thesium chinense* Turcz and Their In Vitro Antioxidant, Anti-Inflammatory and Cytotoxic Activities

**DOI:** 10.3390/molecules28062685

**Published:** 2023-03-16

**Authors:** Zhen-Zhen Liu, Jun-Cheng Ma, Peng Deng, Fu-Cai Ren, Ning Li

**Affiliations:** Inflammation and Immune Mediated Diseases Laboratory of Anhui Province, School of Pharmacy, Anhui Medical University, Hefei 230032, China

**Keywords:** *Thesium chinense* Turcz, chemical constituents, antioxidant, anti-inflammatory activity, cytotoxicity

## Abstract

Three novel compounds (**1–3**) along with twenty-six known compounds, two known steroids (**4–5**) and twenty-four known phenylpropanoids (**6–29**) were isolated from the whole plant of *Thesium chinense* Turcz. The structures of the three new compounds were elucidated on the basis of ESI-MS, HR-ESIMS, 1D and 2D NMR, IR, UV spectroscopic data. The absolute stereochemistry of compound **1** was determined by the Gauge-Including Atomic Orbitals (GIAO) method. The in vitro antioxidant, anti-inflammatory and cytotoxic activities of the isolated compounds were evaluated by DPPH radical-scavenging assay, LPS-activated RAW 264.7 cells model and CCK-8 kit, respectively. Compound **11** showed high antioxidant activity with an SC_50_ value of 16.2 ± 1.6 μM. Compound **21** showed considerable anti-inflammatory activity with an IC_50_ value of 28.6 ± 3.0 μM. Compounds **4** and **5** displayed potent cytotoxic activity against human NCI-H292, SiHa, A549, and MKN45 cell lines, with the compound **4** having IC_50_ values of 17.4 ± 2.4, 22.2 ± 1.1, 9.7 ± 0.9, 9.5 ±0.7 μM, and the compound **5** having all IC_50_ values less than 0.1 μM in vitro.

## 1. Introduction

There are approximately 350 species of the genus *Thesium* (Santalaceae) occurring worldwide in Africa, Europe, Asia, South America and North America [1]. *Thesium chinense* Turcz, a small perennial and hemi-parasitic plant belonging to the family Santalaceae, distributes in East Asia (China, Japan, Korea, and Mongolia) [2]. The whole plant of *T. chinense*, commonly called “Bai-Rui-Cao” in China, was first recorded in the ancient medicinal monograph “Tu Jing Ben Cao” in the North Song Dynasty 1000 years ago. In traditional Chinese medicine (TCM), Bai-Rui-Cao is entitled “Botanical Antibiotics” and is mainly used to treat different diseases including mastitis, pulmonitis, tonsillitis, laryngopharyngitis and upper respiratory tract infections. [3]. Previous studies have reported the presence of polysaccharides, flavonoids, alkaloids, terpenoids, D-mannitol, aromatic compounds and aliphatic acids in the plant of *T. chinense* [4,5]. Modern pharmacological studies have found that *T. chinense* has diverse activities including anti-inflammation [6], antimicrobial effect [7], analgesic activity [8], antioxidant activity [9] and anti-nephropathy [10].

The chemical and pharmacological investigations into *T. chinense* are carried out as part of our ongoing work on the discovery of the bioactive compounds from Chinese medicinal herbs. From EtOAc and n-BuOH extracts of *T. chinense*, three novel compounds (**1–3**), along with twenty-six known compounds, two known steroids (**4–5**) and twenty-four known phenylpropanoids (**6–29**) (Figure 1), are isolated. The structures of the new compounds (**1–3**) are identified on the basis of ESI-MS, HR-ESI-MS, 1D and 2D NMR, IR, UV spectroscopic evidence. To determine the absolute stereochemistry, compound **1** was subjected to the Gauge-Including Atomic Orbitals (GIAO) method. The isolated compounds are evaluated for their antioxidant, anti-inflammatory activities and in vitro cytotoxic activities against four human NCI-H292, SiHa, A549, and MKN45 cancer cell lines by using DPPH radical-scavenging assay, LPS-activated RAW 264.7 cells model and CCK-8 kit, respectively. Herein, we report the isolation, structure determination and activity evaluation, and we preliminarily discuss the structure–activity relationship of the isolates.

## 2. Results and Discussion

### 2.1. Structure Identification

Compound **1** was obtained as a white amorphous powder, and the molecular formula was confirmed as C_25_H_44_O_14_ by HR-ESI-MS (negative) (*m/z* 567.2654 [M−H]^−^, calcd. for C_25_H_43_O_14_, 567.2658). The ^1^H and ^13^C NMR (Table 1) spectrum showed a secondary methyl signal at *δ*_H_ 1.26 (3H, d, *J* = 6.4 Hz, H-10), three tertiary methyl signals at *δ*_H_ 0.87 (3H, s, H-12), *δ*_H_ 1.11 (3H, s, H-13) and *δ*_H_ 1.21 (3H, s, H-11), two methylene signals at *δ*_H_ 1.59 (1H, ddd, *J* = 12.3, 4.3, 2.1 Hz, H-2), *δ*_H_ 1.75 (1H, tt, *J* = 12.2, H-2) and *δ*_H_ 1.87 (1H, m, H-4), *δ*_H_ 1.96 (1H, ddd, *J* = 13.3, 4.3, 2.2 Hz, H-4), two secondary carbinyl proton signals at *δ*_H_ 6.04 (1H, dd, *J* = 15.8, 1.3 Hz, H-7) and *δ*_H_ 5.78 (1H, dd, *J* = 15.8, 6.4 Hz, H-8), two set glucosyl anomeric protons [*δ*_H_ 4.54 (1H, d, *J* = 7.5 Hz, H-1′) and *δ*_H_ 4.54 (1H, d, *J* = 7.5 Hz, H-1″)]. In addition to the signals due to the above functional groups, the spectrum showed signals due to a quaternary carbon atom at 40.8 and two quaternary carbon atoms with an oxygen atom at *δ*_C_ 77.8, 79.1. The side chain structure was confirmed by the ^1^H-^1^H COSY correlations of H-10/H-9, H-9/H-8, and H-8/H-7. Furthermore, according to the HMBC correlations of H-11/C-1, C-2, H-12/C-1, C-6, H-2/C-4, C-6, H-4/C-5, H-13/C-4, and C-6, the ^1^H-^1^H COSY correlations of H-2/H-3, H3/H-4 and the NOESY correlations of H-2 (*δ*_H_ 1.59)/H-3 (*δ*_H_ 4.21), H-11 (*δ*_H_ 1.21)/H-2 (*δ*_H_ 1.59), H-3 (*δ*_H_ 4.21), H-7 (*δ*_H_ 6.04), H-12 (*δ*_H_ 0.87)/H-8 (*δ*_H_ 5.78) and H-13 (*δ*_H_ 1.11)/H-7 (*δ*_H_ 6.04), compound **1** was presumed to have a structure with a megastigm-7-ene carbon skeleton to which four hydroxyl groups are introduced on C-3, 5, 6 and 9. Further in HMBC (Figure 2), the correlation between H-1′ [*δ*_H_ 4.54 (1H, d, *J* = 7.5 Hz)] with C-3 (*δ*_C_ 74.2) suggested that one *β*-glucosyl moiety was located at the C-3 position of the cycle. Correlations of HMBC from H-1″ [*δ*_H_ 4.54 (1H, d, *J* = 7.5 Hz)] with C-2′ (*δ*_C_ 83.7) and H-2′ [*δ*_H_ 3.36 (1H, m)] with C-1″ (*δ*_C_ 105.6) indicated that another *β*-glucosyl moiety was attached at the C-2′ position. To determine the absolute stereochemistry, compound **1** was subjected to the Gauge-Including Atomic Orbitals (GIAO) method [11,12,13]. Compound **1** was represented at the mPW1PW91/6-31G(d) level two isomers **1a** (3R, 5S, 6S, 9R) and **1b** (3S, 5R, 6R, 9S) using Gaussian quantum chemistry NMR calculations and DP4+ probabilistic analysis. The results show that the configuration of **1a** (3R, 5S, 6S, 9R) is more consistent with the experimental value of **1** (Figure 3) and has 99.96% DP4+ probability (Show in Appendix A). Thus, the absolute configuration of compound 1 is confirmed and named as thesiumin A.

Compound **2** was obtained as a white amorphous powder, and the molecular formula was confirmed as C_20_H_30_O_13_ by HR-ESI-MS (negative) (*m/z* 477.1610 [M−H]^−^, calcd. for C_20_H_29_O_13_, 477.1614). The ^1^H NMR spectrum of compound **2** showed the signals of two sets of methylene protons [*δ*_H_ 2.71(2H, t, *J* = 7.1 Hz, H-7) and *δ*_H_ 3.69 (2H, t, *J* = 7.0 Hz, H-8)], an ABX aromatic system proton [*δ*_H_ 7.15(1H, d, *J* = 2.0 Hz, H-2), *δ*_H_ 6.76 (1H, d, *J* = 8.1 Hz, H-5) and *δ*_H_ 6.80 (1H, d, *J* = 2.0, 8.1 Hz, H-6)], and two sets of glucosyl anomeric protons [*δ*_H_ 4.78 (1H, d, *J* = 7.8 Hz, H-1′) and *δ*_H_ 4.74 (1H, d, *J* = 7.8 Hz, H-1″)], which indicated the presence of 3,4- dihydroxyphenyl alcohol and two glucose units. Further comparison of the ^1^H and ^13^C NMR spectra of compound **2** with 3, 4-dihydroxyphenyl alcohol 3-*O*-*β*-D-glucopyranoside (**6**) [14] suggested that compound **2** contained one more glucose unit than **6**. In HMBC (Figure 4), the correlation between H-1′ [*δ*_H_ 4.78 (1H, d, *J* = 7.8 Hz)] with C-3 (*δ*_C_ 146.7) suggested one *β*-glucosyl moiety was located at the C-3 position of the aromatic cycle. Correlations of HMBC from H-1″ [*δ*_H_ 4.74 (1H, d, *J* = 7.8 Hz)] with C-2′ (*δ*_C_ 83.3) and H-2″ [*δ*_H_ 3.75(1H, m)] with C-1″(*δ*_C_ 105.7) indicated that another *β*-glucosyl moiety was attached at the C-2′ position. On the basis of the above evidence, a new phenylethanoid glucoside was structurally determined as 3, 4-dihydroxyphenethyl alcohol 3-*O*-β-D-glucopyranosyl (1→2)-β-D-glucopyranoside (**2**), named thesiumin B.

Compound **3** was obtained as a white amorphous powder, and the molecular formula was confirmed as C_18_H_32_O_13_ by HR-ESI-MS (negative) (*m*/*z* 423.1871 [M−H]^−^, calcd. for C_18_H_31_O_13_, 423.1874). The ^1^H and ^13^C NMR (Table 2) together with the HSQC spectrum showed three sets of methylenes [*δ*_H_ 2.00 (2H, m, *J* = 7.3 Hz), *δ*_C_ 20.3, C-5; *δ*_H_ 3.44 (2H, m), *δ*_C_ 27.6, C-2; *δ*_H_ 3.74 (2H, m), *δ*_C_ 68.3, C-1], one methyl group [*δ*_H_ 0.91 (3H, t, *J* = 7.5 Hz), *δ*_C_ 14.3, C-6], one cis-configuration double bond [*δ*_H_ 5.34 (1H, dt, *J* = 11.3, 6.9 Hz), *δ*_C_ 125.4, C-3; *δ*_H_ 5.39 (1H, dt, *J* = 11.3, 6.9 Hz), *δ*_C_ 132.9, C-4], and two sets of glucosyl anomeric protons [*δ*_H_ 4.29 (1H, d, *J* = 7.7 Hz,), *δ*_C_ 101.4, C-1′; *δ*_H_ 4.37 (1H, d, *J* = 7.7 Hz), *δ*_C_ 104.2, C-1″], which indicated the presence of a carbon chain and two glucose units. In HMBC (Figure 4), the correlation between H-1′ [*δ*_H_ 4.29 (1H, d, *J* = 7.7 Hz)] with C-1 (*δ*_C_ 68.3) suggested that one *β*-glucosyl moiety was located at the C-1 position of the carbon chain. Correlations of HMBC from H-1″ [*δ*_H_ 4.29 (1H, d, *J* = 7.7 Hz)] with C-2′ (*δ*_C_ 82.5) and H-2″ [*δ*_H_ 2.98(1H, t, *J* = 8.2 Hz)] with C-1″(*δ*_C_ 104.2) indicated that another *β*-glucosyl moiety was attached at the C-2′ position. Furthermore, in HMBC, there was correlation between H-1 with C-3, and the ^1^H-^1^H COSY correlations of H-2/H-3, and H-4/H-5. Thus, compound **3** was identified as shown and named as (Z)-hex-3-ene 3-*O*-β-D- glucopyranosyl (1→2)-β-D- glucopyranoside, named thesiumin C.

Beside the three novel compounds (**1–3**), twenty-six known compounds (**4–29**), including two steroids (**4–5**) and twenty-four phenylpropanoids (**6–29**), were also identified from the whole plants of *T. chinense*. Their strctures were determined as: perlplogemn [15] (**4**), periplocin [16] (**5**), 2-hydroxy-5-(2-hydroxy-ethyl) phenyl-*β*-D-glucopyranoside (**6**) [11], 3, 5-dihydroxyphenethyl alcohol 3-*O*-*β*-D-glucopyranoside (**7**) [17], phenethyl alcohol *β*-sophoroside (**8**) [18], samsesquinoside (**9**) [19], pinoresinol-4-*O*-D- glucoside (**10**) [20], syringaresinol-4′-*O*-*β*-D-glucopyranoside (**11**) [21], dihydrodehydro diconiferyl alcohol 4-*O*-*β*-D-glucoside (**12**) [22], (7S, 8R) 9′- methoxy- dehydrodiconiferyl alcohol 4-*O*-β-D-glucopyranoside (**13**) [23], picraquassioside C (**14**) [24], citrusin A (**15**) [25], citrusin B (**16**) [25], (erythro, erythro)-1-[4-[2-hydroxy-2-(4-hydroxy-3-methoxy-phenyl)-1-(hydroxy methyl) ethoxy]-3, 5-dimethoxy phenyl]-1, 2, 3-propanetriol (**17**) [26], 1-(4-hydroxy-3-methoxy)-phenyl -2-[4-(1, 2, 3-trihydroxy propyl)-2-methoxy]-phenoxy-1, 3-propan-diol (**18**) [27], lariciresinol-4-*O*-*β*-D-glucoside (**19**) [28], ficusal (**20**) [29], 3-*O*-ethyl ferulate (**21**) [30], *p*-coumaric acid (**22**) [31], feruloylquinic acid (**23**) [32], syringin (**24**) [33], coniferin (**25**) [34], alatusol D (**26**) [35], scopolin (**27**) [36], scopoletin (**28**) [36] and sinapaldehyde glucoside (**29**) [37]. All the known compounds (**4–29**) were isolated from *Thesium chinense* Turcz for the first time.

### 2.2. Bio-Activities of Compounds **2–29**


#### 2.2.1. Antioxidant Activity

The antioxidant activities of all the isolated compounds except compound **1** were evaluated using DPPH radical scavenging assay (Table 3). Comparing with positive control (ascorbic acid, SC_50_ = 15.5 ± 0.8 μM), compound **11** showed high antioxidant activity with an SC_50_ value of 16.2 ± 1.6 μM. Compounds **9**, **10**, **14**, **17**, **21** and **23** exhibited moderate activities with SC_50_ values ranging from 30.5 to 75.6 μM. Compounds **6**, **7**, **18** and **26** displayed weak antioxidant activities on radical scavenging.

#### 2.2.2. Anti-Inflammatory Activity

The in vitro anti-inflammatory activities of compounds **2–29** were evaluated by their inhibitory effects on NO production in LPS-activated RAW 264.7 cell models, and the results showed that most compounds had no anti-inflammatory activities except for ethyl ferulate (**21**) (Table 4). Compared with the positive control quercetin (IC_50_ = 11.1 ± 1.4 μM), compound **21** displayed considerable anti-inflammatory activity with an IC_50_ value of 28.6 ± 3.0 μM. Recent in vivo studies reported that ethyl ferulate (**21**) also displayed obvious anti-inflammatory effects against LPS-induced acute lung injury in mice [38,39].

#### 2.2.3. Cytotoxic Activity

In the CCK-8 assay, many of the isolated compounds from *T. chinense* showed moderate or considerable cytotoxic activities against four human cancer cell lines of A549, NCI-H292, SiHa and MKN45 (Table 5). Among them, Scopoletin (**28**) showed moderate and selective cytotoxic activity against MKN45 with the IC_50_ value of 52.8 ± 5.3 μM. Perlplogemn (**4**) displayed potent cytotoxic activity against NCI-H292, SiHa, A549, and MKN45 with IC_50_ values of 17.4 ± 2.4, 22.2 ± 1.1, 9.7 ± 0.9, and 9.5 ±0.7 μM, respectively. It is worth noting that periplocin (**5**) demonstrated promising cytotoxic activity against the four human cancer cell lines with all the IC_50_ values less than 0.1 μM compared to that of the positive control cisplatin (IC_50_ = 7.0 ± 0.3 μM). Both perlplogemn (**4**) and periplocin (**5**) are known for their cytotoxicity towards different cancer cells, and are now considered as potent leading compounds for several tumors with high drug resistance [40,41,42].

## 3. Materials and Methods

### 3.1. General Experimental Procedures

A Shimadzu UV-2401PC spectrophotometer was used to obtain the UV spectra. A Thermo NICOLET Is10 FT-IR spectrometer was used for IR spectra with KBr pellets. 1D and 2D NMR spectra were recorded on an Avance III-600 spectrometer with TMS as internal standard, and chemical shifts (*δ*) are expressed in ppm. MS and HR-MS were performed on an Agilent 1290 UPLC/6540 Q-TOF spectrometer. Column chromatography was carried out on Sephadex LH-20 gel (25–100 μm, Pharmacia Fine Chemical Co. Ltd., Stockholm, Sweden), MCI-gel (75–150 µm, Mitsubishi Chemical Corporation, Tokyo, Japan), ODS silica gel (50 µm, YMC Ltd., Kyoto, Japan) and silica gel (200–300 mesh, Qingdao Haiyang Chemical Co. Ltd., Qingdao, China). Thin layer chromatography (TLC) was carried out on silica gel GF254 precoated plates (Qingdao Haiyang Chemical Co. Ltd., Qingdao, China), and spots were detected by spraying with 5% H_2_SO_4_ in EtOH followed by heating.

### 3.2. Plant Material

The whole dry herb of *T. chinense* was collected from Xiangyang city, Hubei Province, China in May 2019. The species was identified by Prof. Kai-Jin Wang at the School of Life Sciences, Anhui University, and a voucher specimen (No. 20190927) was deposited in the School of Pharmacy, Anhui Medical University.

### 3.3. Extraction and Isolation

The air-dried whole plant of *T. chinense* (10 kg) was extracted with 85% EtOH (3 × 100 L, each 4 h) at 60 °C. The combined EtOH extracts were evaporated at 60 °C using a rotatory evaporator with a vacuum pump to obtain suspended water (9 L), the suspension was successively extracted with petroleum ether, EtOAc, n-BuOH (1:2, *v*/*v*, three times each).

The EtOAc fraction (369 g) was partitioned into six fractions (Fr. E1→E6) by MCI gel column chromatography (CC) eluted with EtOH-H_2_O (50:50 to 100:0, *v*/*v*). Fr. E1 was separated by MCI gel CC eluted with MeOH-H_2_O (10:90 to 100:0, *v*/*v*) to afford five subfractions (Fr. E1-1→Fr. E1-5). Fr. E1-4 was fractioned over Sephadex LH-20 gel CC eluted with MeOH-H_2_O (10:90 to 100:0, *v*/*v*) and further purified by silica gel CC eluted with CH_2_Cl_2_-MeOH (40:1) to obtain compounds **28** (25.0 mg) and **29** (5.0 mg). Fr. E2 was chromatographed on silica gel CC eluted with a gradient of CH_2_Cl_2_-MeOH (100:1 to 50:1, *v*/*v*) and then purified by ODS gel CC eluted with MeOH-H_2_O (5:95 to 100:0, *v*/*v*) to yield compound **20** (4.0 mg). Fr. E3 was subjected to Sephadex LH-20 gel CC eluted with EtOH-H_2_O (50:50, *v*/*v*) and then purified by ODS gel CC to obtain compound **21**(4.3 mg).

The n-BuOH fraction (950 g) was separated by silica gel CC eluted with CH_2_Cl_2_-MeOH-H_2_O (12.25:3:0.1, *v*/*v*) to afford four fractions (Fr.1→4). Fr.1 was further separated by silica gel CC eluted with a gradient of CH_2_Cl_2_-MeOH (25:1 to 1:1, *v*/*v*) to produce three fractions (Fr.1-1→Fr.1-3). Fr.1-1 was subjected to MCI gel CC eluted with MeOH-H_2_O (10:90 to 100:0, *v*/*v*) to afford six subfractions (Fr.1-1-1→Fr.1-1-6). Fr.1-1-3 was separated by Sephadex LH-20 gel CC and then purified by silica gel CC eluted with a gradient of CH_2_Cl_2_-MeOH (50:1 to 18:1, *v*/*v*) to give compound **3** (63 mg) and **26** (7.8 mg), and further purified by ODS gel CC eluted with MeOH-H_2_O (5:95 to 100:0, *v*/*v*) to obtain compounds **17** (10.3 mg), **27** (2.2 mg) and **23** (15.0 mg). Fr.1-1-5 was chromatographed on Sephadex LH-20 gel CC eluted with MeOH (10:90, *v*/*v*) and further purified by ODS gel CC eluted with MeOH-H_2_O (5:95, *v*/*v*) to yield compounds **1** (1.8 mg), **9** (10.0 mg), **10** (25.0 mg) and **11** (25.0 mg). Fr.1-2 was fractioned on MCI gel CC eluted with a gradient of MeOH-H_2_O (10:90 to 100:0, *v*/*v*) to produce eleven fractions (Fr.1-2-1→Fr.1-2-5). Fr.1-2-2 was subjected to Sephadex LH-20 gel CC and further purified by ODS gel CC eluted with MeOH-H_2_O (5:95, *v*/*v*) to obtain compounds **18** (10.0 mg), **24** (2.3 mg) and **25** (6.0 mg), respectively. Fr.1-2-5 was chromatographed on Sephadex LH-20 gel CC eluted with MeOH-H_2_O (10:90, *v*/*v*) and further purified by ODS gel CC eluted with MeOH-H_2_O (5:95, *v*/*v*) to give compounds **13** (77.0 mg), **19** (120.0 mg), **16** (20.0 mg), **14** (2.3 mg) and **22** (15.5 mg). Fr.1-3 was fractioned over Sephadex LH-20 gel CC eluted with MeOH-H_2_O (10:90, *v*/*v*) and further purified by silica gel CC eluted with CH_2_Cl_2_-MeOH (15:1, *v*/*v*) to obtain compounds **4** (86 mg), **12** (220.0 mg) and **7** (80.0 mg). Fr.2 was subjected to silica gel CC eluted with CH_2_Cl_2_-MeOH (5:1, *v*/*v*) to give three fractions (Fr.2-1→Fr.2-3). Fr.2-3 was fractioned over Sephadex LH-20 gel CC eluted with a gradient of MeOH-H_2_O (10:90 to 100:0, *v*/*v*) and then purified by recrystallization to obtain compounds **15** (40.0 mg) and **8** (14.0 mg). Fr.3 was chromatographed on silica gel CC eluted with CH_2_Cl_2_-MeOH (16:1 to 2:1, *v*/*v*) and separated by Sephadex LH-20 gel CC eluted with MeOH-H_2_O (10:90, *v*/*v*) to yield compounds **5** (96 mg) and **6** (210.0 mg), and further purified by ODS gel CC eluted with MeOH-H_2_O (10:90, *v*/*v*) to obtain compound **2** (66.0 mg).

#### Characterization of the Isolated Compounds **1**, **2**, **3**:

Thesiumin A (**1**): White amorphous powder; [α]${{}_{D}^{25}}_{}^{}$ = −26.70 (*c* 0.10, MeOH); UV (MeOH) *λ*_max_ (log *ε*): 204 (3.4) nm; IR (KBr) *ν*_max_: 3391, 2960, 2923, 2876, 2854, 1735, 1606, 1508, 1456, 1383, 1317, 1262, 1229, 1171, 1074, 1030, 938, 895, 862, 803, 774, 747, 718, 623, 579 cm^−1^; ESI-MS (negative) *m/z*: 568 [M−H]^−^, HR-ESI-MS (negative) *m/z* 567.2654 [M−H]^−^ (calcd. for C_25_H_43_O_14_, 567.2658).

Thesiumin B (**2**): White amorphous powder; [α]${{}_{D}^{25}}_{}^{}$ = −1.20 (*c* 0.30, MeOH); UV (MeOH) *λ*_max_ (log *ε*): 202.5 (4.28), 217.5 (3.95), 279.0 (3.50) nm; IR (KBr) *ν*_max_ 3456, 3390, 2927, 2882, 1608, 1512, 1436, 1382, 1374, 1274, 1233, 1134, 1077, 885, 825, 804, 565 cm^−1^; ESI-MS (negative) *m/z*: 477 [M−H]^−^, HR-ESI-MS (negative) *m/z* 477.1610 [M−H]^−^ (calcd. for C_20_H_29_O_13_, 477.1614).

Thesiumin C (**3**): White amorphous powder; [α]${{}_{D}^{25}}_{}^{}$ = −14.22 (*c* 1.00, DMSO); UV (DMSO) *λ*_max_ (log *ε*): 252.5 (1.1), 279.5 (1.4) nm; IR (KBr) *ν*_max_: 3371, 3008, 2963, 2964, 2881, 1409, 1371, 1231, 1161, 1077, 895 cm^−1^; ESI-MS (negative) *m/z*: 424 [M−H]^−^, HR-ESI-MS (negative) *m/z* 423.1871 [M−H]^−^ (calcd. for C_18_H_31_O_13_, 423.1874).

### 3.4. DPPH Radical Scavenging Assay

The 2, 2-diphenyl-1-picrylhydrazyl (Aldrich Chem. Co. Ltd, Shanghai, China) radical scavenging activity assay was performed according to the previously published method [43] with some modification. Test samples were dissolved in MeOH to six different concentrations ranging from 3.125 to 100 μM. Then, 100 μL of 100 μM DPPH in MeOH were mixed with 100 μL of samples at different concentrations in the wells of 96-well plates. The reaction mixtures were incubated in the dark at 37 °C for 30 min and then measured at 517 nm. Ascorbic acid was used as the positive control. The scavenging activity (SC) was estimated as follows: % SC = [1 − (A_sample_ − A_blank_)/A_control_] ×100. A_sample_ was the equivalent mixture of test sample and DPPH solution. A_blank_ was the equivalent mixture of test sample and MeOH solution. A_control_ was the equivalent mixture of DPPH and MeOH solution. The SC_50_ values were used to evaluate the antioxidant activities of isolated compounds. The experiments were performed in triplicate, and the data were expressed as the means ± SD of three independent experiments.

### 3.5. Anti-Inflammatory Activity 

The macrophage RAW 264.7 cells were obtained from the Cell Bank of the Chinese Academy of Sciences (Shanghai, China). A total of 180 μL of RAW 264.7 cells (1 × 10^4^) were plated respectively in each well of 96-well plate and cultured for 24 h, then the supernatant of the culture was discarded and 180 μL complete medium with 1 μg/mL LPS and different concentrations of test samples were added into triplicate wells for 24 h. The inhibition effects on LPS-stimulated NO production was evaluated by the Griess reaction assay. The experiments were performed in triplicate, and the data were expressed as the means ± SD of three independent experiments.

The cell viability was determined by the CCK-8 assay method. The cytotoxicity was calculated from the plotted results using untreated cells at 100%.

### 3.6. Cytotoxic Assay (CCK-8 Assay) 

Four human cancer cell lines, A549, NCI-H292, SiHa and MKN45, were purchased from the Cell Bank of the Chinese Academy of Sciences (Shanghai, China). All cancer cells were cultured in 1640 medium (Thermo Fisher Scientific, Waltham, MA, USA) supplemented with 10% FBS (Sijiqing, Huzhou, China) at 37 °C in a humidified atmosphere with 5% CO_2_. Cisplatin (Abmole Bioscience, Shanghai, China) was used as positive control.

A total of 180 μL of cancer cells (5 × 10^3^) were plated in each well of 96-well plates and cultured for 24 h, then the supernatant of the culture was discarded and 180 μL complete medium with different concentrations of test samples were added into triplicate wells of each cell line, respectively. Thereafter, A549, NCI-H292, SiHa and MKN45 were incubated for 48 h, respectively. The cell viability was evaluated by Cell Counting Kit-8 (CCK-8, Best Bio., Beijing, China), respectively. The experiments were performed in triplicate, and the data were expressed as the means ± SD of three independent experiments.

## 4. Conclusions

In summary, three new compounds (**1–3**) along with two known steroids (**4–5**) and twenty-four known phenylpropanoids (**6–29**) were isolated from the EtOAc and n-BuOH extracts from the whole plant of *T. chinense.* All the compounds were isolated from this species for the first time. The isolates except for compound **1** are evaluated for their antioxidant and anti-inflammatory activities as well as their in vitro cytotoxic activities against four human cancer cell lines. The two new biglycosides (**2** and **3**) have no activity. Only compound **21** displayed considerable anti-inflammatory activity with an IC_50_ value of 28.6 ± 3.0 μM by inhibited LPS-induced NO production in RAW 264.7 cells. Most phenylpropanoids with a phenolic hydroxyl group displayed significant or moderate scavenging activity on DPPH radicals (Table 3). Compared to the positive control ascorbic acid, the three furofuran lignan glycosides (**9–11**) and tetrahydrofuran lignan glycoside (**19**) displayed significant scavenging activity on DPPH radicals. The two oxyneoligan **17** and **18** showed moderate activity, and compound **17** displayed much stronger activity than that of **18**, which indicated that the methoxy substitution on the benzene ring could enhance the activity. Two phenylethyl derivatives **6** and **7** exhibited weak activity. The phenylpropanoids **8**, **12–16** and **29** without phenolic hydroxyl group did not show activity, which suggests that the phenolic hydroxyl group may be one of the key factors in the enhancement of the antioxidant activity.

Two steroid compounds **4** and **5** with α,β-unsaturated lactone ring showed extremely strong cytotoxicity against all four tested cancer cell lines, with IC_50_ values ranging from < 0.1 to 22.2 ± 1.1 μM, relative to the positive control, cisplatin (IC_50_ values from 7.0 ± 0.3 to 52.8 ± 5.3 μM). In particular, compound **5** showed the strongest cytotoxicity, and its four IC_50_ values were less than 0.1 μM. Compound **5** was formed by *O*-glycosylation at C-3 of **4**, which allowed us to reach a preliminary deduction that *O*-glycosylation at C-3 caused a more positive effect on cytotoxic activity. Compound **28** showed highly selective cytotoxic activity against MKN45 with the IC_50_ value of 52.8 ± 5.3 μM among the four human cancer cell lines. Of all the compounds tested, only compounds **4**, **5** and **28,** with *α*, *β*-unsaturated lactone ring, showed certain cytotoxicity, which suggested that *α*, *β*-unsaturated lactone ring was an active functional group of anti-tumor, and this research result is consistent with the literature report [44]. 

This study not only revealed the structural diversity of the chemical components of *T. chinense*, but also provided lead compounds for the development of antioxidants, anti-inflammatory and antitumor agents, and promoted reasonable use of this herb.

## Figures and Tables

**Figure 1 molecules-28-02685-f001:**
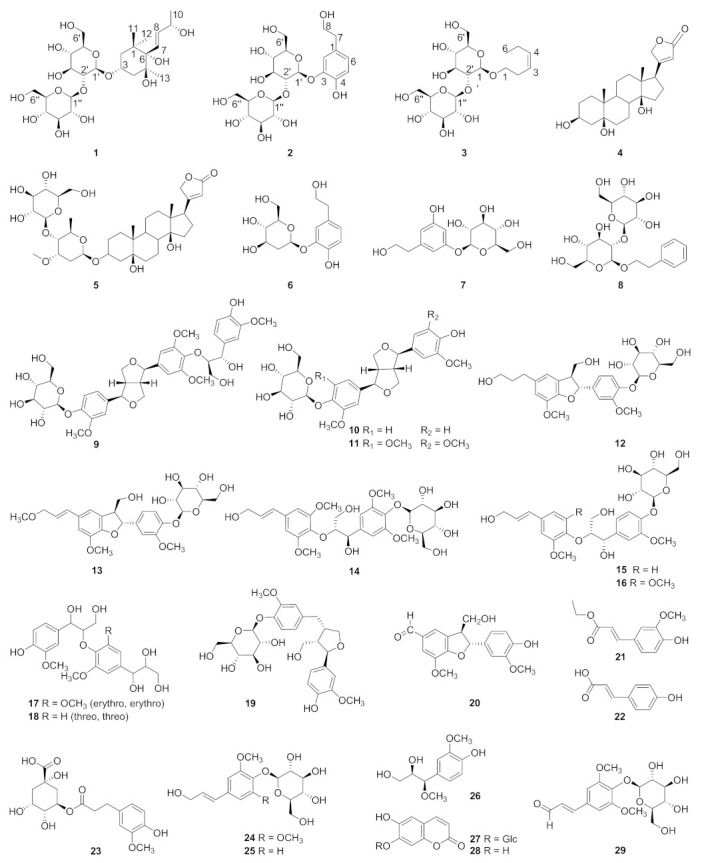
Chemical structures of compounds **1–29**.

**Figure 2 molecules-28-02685-f002:**
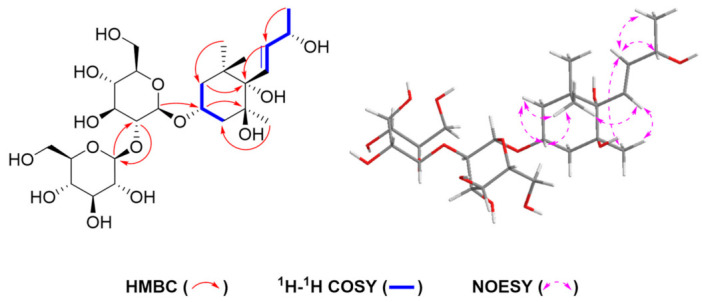
Selected HMBC, ^1^H-^1^H COSY and NOSY correlations of **1**.

**Figure 3 molecules-28-02685-f003:**
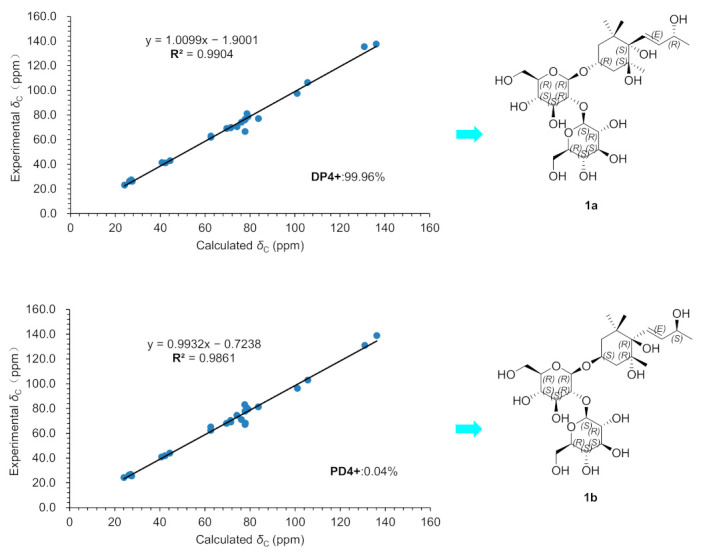
The ^13^C NMR calculations of **1a** and **1b**.

**Figure 4 molecules-28-02685-f004:**
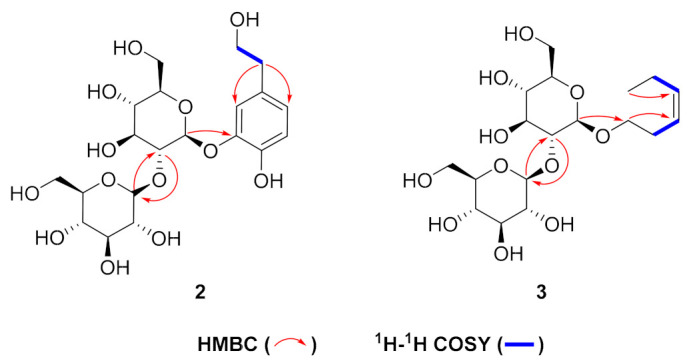
Selected HMBC and ^1^H-^1^H COSY correlations of **2** and **3**.

**Table 1 molecules-28-02685-t001:** ^1^H (800 MHz) and ^13^C (200 MHz) NMR data of compound **1** in CD_3_OD, (*δ* in ppm, *J* in Hz).

Position	*δ* _H_	*δ* _C_	Position	*δ* _H_	*δ* _C_
1	-	40.8	1′	4.54 (d, *J* = 7.5 Hz)	101.0
2	1.59 (ddd, *J* = 12.3, 4.3, 2.1 Hz) 1.75 (t, *J* = 12.2 Hz)	44.4	2′	3.36 (m)	83.7
3	4.21 (tt, *J* = 11.6, 4.3 Hz)	74.2	3′	3.54 (m)	77.8
4	1.87 (m) 1.96 (ddd, *J* = 13.3, 4.3, 2.2 Hz)	42.3	4′	3.33 (m)	71.4
5	-	77.8	5′	3.28 (m)	78.6
6	-	79.1	6′	3.68 (m) 3.85 (dd, *J* = 12.0, 2.3 Hz)	62.6
7	6.04 (dd, *J* = 15.8, 1.3 Hz)	130.9	1″	4.54 (d, *J* = 7.5 Hz)	105.6
8	5.78 (dd, *J* = 15.8, 6.4 Hz)	136.2	2″	3.24 (m)	76.2
9	4.33 (d, *J* = 6.4 Hz)	69.6	3″	3.27 (m)	77.9
10	1.26 (d, *J* = 6.4 Hz)	24.1	4″	3.28 (m)	71.5
11	1.21 (s)	26.3	5″	3.37 (m)	77.7
12	0.87 (s)	27.5	6″	3.68 (m) 3.92 (dd, *J* = 12.1, 1.6 Hz)	62.6
13	1.11 (s)	27.1			

**Table 2 molecules-28-02685-t002:** ^1^H (600 MHz) and ^13^C (150 MHz) NMR data of compounds **2** in CD_3_OD, **3** in DMSO (*δ* in ppm, *J* in Hz).

Position	2		3	
	*δ* _H_	*δ* _C_	*δ* _H_	*δ* _C_
1	-	132.1	3.74 (m) 3.44 (m)	68.3
2	7.15 (d, *J* = 2.0 Hz)	119.9	2.26 (m)	27.6
3	-	146.9	5.34 (dt, *J* = 11.3, 6.9 Hz)	125.4
4	-	146.7	5.39 (dt, *J* = 11.1, 6.9 Hz)	132.9
5	6.76 (d, *J* = 8.1 Hz)	116.4	2.00 (m, *J* = 7.3 Hz)	20.3
6	6.80 (dd, *J* = 2.0, 8.1 Hz)	125.6	0.91 (t, *J* = 7.5 Hz)	14.3
7	2.72 (t, *J* = 7.1 Hz)	39.5	-	-
8	3.69 (t, *J* = 7.0 Hz)	64.3	-	-
1′	4.78 (d, *J* = 7.8 Hz)	103.6	4.29 (d, *J* = 7.7 Hz)	101.4
2′	3.75 (m)	83.3	3.20 (t, *J* = 8.4 Hz)	82.5
3′	3.40 (m)	78.2	3.07 (m)	77.1
4′	3.43 (m)	71.0	3.10 (m)	69.9
5′	3.64 (t, *J* = 8.9 Hz)	77.8	3.35 (t, *J* = 8.5 Hz)	76.1
6′	3.77 (m) 3.71 (m)	62.3	3.62 (m) 3.50 (m)	60.9
1″	4.74 (d, *J* = 7.8 Hz)	105.7	4.37 (d, *J* = 8.1 Hz)	104.2
2″	3.28 (dd, *J* = 9.2, 7.9 Hz)	75.8	2.98 (t, *J* = 8.2 Hz)	75.0
3″	3.36 (m)	78.5	3.12 (m)	76.7
4″	3.35 (m)	71.3	3.10 (m)	69.8
5″	3.74 (m)	77.7	3.13 (d, *J* = 2.6 Hz)	76.2
6″	3.93 (m) 3.70 (m)	64.3	3.66 (m) 3.43 (m)	61.0

**Table 3 molecules-28-02685-t003:** Antioxidant activities of compounds and positive control in DPPH radical scavenging assay.

Compound	SC_50_ (μM)	Compound	SC_50_ (μM)
**1**	nd *	**16**	nc **
**2**	nc **	**17**	75.6 ± 6.1
**3**	nc **	**18**	127.9 ± 15.4
**4**	nc **	**19**	48.8 ± 2.9
**5**	nc **	**20**	nc **
**6**	194.9 ± 12.6	**21**	50.8 ± 1.4
**7**	255.2 ± 12.6	**22**	nc **
**8**	nc **	**23**	36.4 ± 5.0
**9**	47.3 ± 2.3	**24**	50.8 ± 1.4
**10**	30.5 ± 6.5	**25**	nc **
**11**	16.2 ± 1.6	**26**	284.0 ± 16.5
**12**	nc **	**27**	nc **
**13**	nc **	**28**	nc **
**14**	nc **	**29**	nc **
**15**	nc **	Ascorbic acid	15.6 ± 0.8

* nd, the weight was too small to conduct related detect. ** nc, SC50 cannot be calculated by GraphPad Prism 9.0 software due to weak or no related activities. SC50: radical-scavenging activity (concentration in μM required for 50% reduction of DPPH radicals).

**Table 4 molecules-28-02685-t004:** NO inhibitory activities of compounds and positive control in RAW 264.7 cell line.

Compounds	IC_50_ (μM)	Cytotoxicity (IC_50_)	Compounds	IC_50_ (μM)	Cytotoxicity (IC_50_)
**1**	nd *	nd *	**2–20**, **22–29**	>200	>200
**21**	28.6 ± 3.0	>200	Quercetin	11.1 ± 1.4	>50

* nd, the amount of compound **1** was too small for activity detection. IC_50_ values were expressed as mean ± SD (*n* = 3).

**Table 5 molecules-28-02685-t005:** Human Cancer Cell Proliferation Inhibition of Compounds and Positive Control.

Compounds	IC_50_ (μM)			
	A549	NCI-H292	SiHa	MKN45
**1**	nd *	nd *	nd *	nd *
**4**	17.4 ± 2.4	22.2 ± 1.1	9.7 ± 0.9	9.5 ±0.7
**5**	0.1>	0.1>	0.1>	0.1>
**28**	>200	>200	>200	52.8 ± 5.3
**2, 3, 6–27, 29**	>200	>200	>200	>200
**cisplatin**	19.5 ± 6.7	52.8 ± 5.3	7.0 ± 0.7	7.0 ± 0.3

* nd, the amount of compound **1** was too small for activity detection. IC_50_ values were expressed as mean ± SD (*n* = 3).

## Data Availability

The data presented in this study are available on request from the corresponding authors.

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
