# Peer review of "Chemical Constituents of Thesium chinense Turcz and Their In Vitro Antioxidant, Anti-Inflammatory and Cytotoxic Activities"

_molecules, 2023, doi:10.3390/molecules28062685_

Round 1
Reviewer 1 Report
· General: although the structure elucidation for compounds is good, I found the in vitro studies not clear nor focused. There is no clear discussion why the compounds were scanned for Antioxidant, Anti-Inflammatory and Cytotoxic Activities in specific. Why these cell lines has been selected for canning. All in vitro results are not clearly displayed.
· The identification for compounds 1, 2 & 3 are very clear, well comprehensive written
· Page 6: last line in the 1st paragraph: “All the known compounds (4-29) were isolated from Thesium chinense Turcz for the first time”: although the compounds are not novel, such claim for isolation these compounds for the 1st time in this herb need further evidencing. Please write how did the researchers
· Page 7: for Anti-inflammatory activity: there is no table or figures for the results, please add one. OR if the data are not significant, You can modify the article title as it state the Anti-inflammatory activity. Thus the title is misleading. Why compound 1 was not scanned for its Anti-inflammatory effect?
· Page 7: Cytotoxic activity: must add graph or table clearly displaying the results
Author Response
General: although the structure elucidation for compounds is good, I found the in vitro studies not clear nor focused. There is no clear discussion why the compounds were scanned for Antioxidant, Anti-Inflammatory and Cytotoxic Activities in specific. Why these cell lines have been selected for canning. All in vitro results are not clearly displayed.
Response: We appreciate this comment very much, which have contributed to an improvement of this paper. In traditional Chinese medicine, Thesium chinense Turcz is entitled “Botanical Antibiotics” and mainly used to treat inflammatory related diseases and upper respiratory tract infections, so we scanned the isolates for anti-Inflammatory and cytotoxic activities associated with lung cancer (NCI-H292 and A549). SiHa and MKN45 cell lines were selected in order to further explore the activity of isolates in other tumor cell lines. Most of the isolated compounds contain phenolic hydroxyl groups, so the isolates were designed for their antioxidant activity. In order to display the in vitro results more clearly, we added Table 4 and Table 5.
- Page 6: last line in the 1st paragraph: “All the known compounds (4-29) were isolated from Thesium chinense Turcz for the first time”: although the compounds are not novel, such claim for isolation these compounds for the 1st time in this herb need further evidencing. Please write how did the researchers.
Response: Thank you very much for your professional question. We have introduced the types of chemical components reported in the literature in line 36-37 in Introduction, mainly including polysaccharides, flavonoids, alkaloids, terpenoids, D-mannitol, aromatic compounds and aliphatic acids from the plant of T. chinense. However, two known steroids (4-5) and twenty-four known phenylpropanoids (6-29) isolated from this species in this paper have not been reported in the literature.
- Page 7: for Anti-inflammatory activity: there is no table or figures for the results, please add one. OR if the data are not significant, you can modify the article title as it states the Anti-inflammatory activity. Thus, the title is misleading. Why compound 1 was not scanned for its Anti-inflammatory effect?
Response: Thank you very much for your professional question. We have added the anti-inflammatory results in Table 4 in the revision. We were not scanned for anti-inflammatory effect of compound 1 in that the weight compound 1 (just 1.8 mg) was too small to do related experiments after structural identification. The details are described below Table 4.
The comment resulted adds lines 152-154 in the revision.
- Page 7: Cytotoxic activity: must add graph or table clearly displaying the results.
Response: Thank you very much for your kindly suggestion, and we have added the cytotoxic results in Table 5.
The comment resulted adds lines 167-169 in the revision.
Reviewer 2 Report
Even though the manuscript “Chemical Constituents of Thesium chinense Turcz and Their in Vitro Antioxidant, Anti-Inflammatory and Cytotoxic Activities” may be of interest to the Molecules readers, I believe that major changes should be done to accept it for publication. In general terms, the discussion should be improved.
Some specific recommendations are enlisted below:
The introduction should be revised, the last part resembles an abstract. Authors should describe the relevance of carrying studies in which the structure-activity relationship (SAR) is considered because this is the main objective of the study.
Authors should describe in the text where the supplementary material should be revised. They are presenting several NMR spectra, MS values, and statistical analyses which are not described or discussed in the manuscript
Discussion of Section 2.2 should be improved (all subsections). In section 2.2.1 Antioxidant activity, authors are only describing their results, without discussing the relevance of changes in the structure of the different molecules. In all cases the SAR should be discussed.
It is not clear why compound 1 (new compound) was not evaluated. In table 3, it is not clear the difference between not detected (nd) and not calculated (nc).
I believe that results from sections 2.2.2 and 2.2.3 could be presented in a table.
Author Response
- Even though the manuscript “Chemical Constituents of Thesium chinense Turcz and Their in Vitro Antioxidant, Anti-Inflammatory and Cytotoxic Activities” may be of interest to the Molecules readers, I believe that major changes should be done to accept it for publication. In general terms, the discussion should be improved.
Response: Thank you very much for your professional advice, we have discussed the structure-activity relationship of compounds. The comment resulted changes in Conclusions in the revision.
- The introduction should be revised, the last part resembles an abstract. Authors should describe the relevance of carrying studies in which the structure-activity relationship (SAR) is considered because this is the main objective of the study.
Response: We appreciate this comment very much, which have contributed to an improvement of this paper. We have revised the introduction. The comment resulted changes in lines 49 and 50 in Introduction.
- Authors should describe in the text where the supplementary material should be revised. They are presenting several NMR spectra, MS values, and statistical analyses which are not described or discussed in the manuscript.
Response: Thank you very much for your carefully review and kindly suggestion, and the relevant statistical analyses were displayed table 1 (line 80), table 2 (117) and last paragraph of 3.3. Extraction and isolation (line 187)
The comment resulted adds in Characterization of the Isolated Compounds 1, 2, 3 (line 225) in the revision.
- Discussion of Section 2.2 should be improved (all subsections). In section 2.2.1 Antioxidant activity, authors are only describing their results, without discussing the relevance of changes in the structure of the different molecules. In all cases the SAR should be discussed.
Response: Thank you very much for your professional advice. We have added the discussed in this version.
The comment resulted changes in Conclusion (line 274-285) in the revision.
- It is not clear why compound 1 (new compound) was not evaluated. In table 3, it is not clear the difference between not detected (nd) and not calculated (nc).
Response: Thank you. We were not scanned for these activities of compound 1 in that the weight of compound 1 (just 1.8 mg) was too small to do related experiments after structural identification. nc: IC50 cannot be calculated by GraphPad Prism 9.0 software due to weak or no related activities. nd: The weight of isolated compound was too small to be detected. We have specified the meaning of nd and nc below Table 3-5.
- I believe that results from sections 2.2.2 and 2.2.3 could be presented in a table.
Response: Thank you very much for your kindly suggestion, and we have added anti-inflammatory results in Table 4 and the cytotoxic results in Table 5.
The comment resulted adds Table 4 and Table 5 in the revision.
Reviewer 3 Report
The authors have described in detail isolation methods for isolation of chemical constituens of Thesium chinense Turcz. They have also tested antioxidant, anti-inflammatory and cytotoxic effects of isolated compound after identifing their structure.
The obtained results are very scientifically important and the manuscript is well written.
However, there is a list of several suggestions, comments and questions to authors
1. The second paragraph in Introduction section could be rewritten as the authors do not describe literature data, but the plan sheme of their experiment.
They could add some data about presence of isolated molecules from Thesium chinense in other plant species, literature data about previous studies about active principes in Thesium chin, or similiar data...
2. The authors could explain why did not test antioxidant, anti-inflamnatory and citotoxic effects of supstance 1?
Author Response
- The second paragraph in Introduction section could be rewritten as the authors do not describe literature data, but the plan sheme of their experiment.
They could add some data about presence of isolated molecules from Thesium chinense in other plant species, literature data about previous studies about active principes in Thesium chin, or similiar data...
Response: We appreciate this comment very much. We have described the main chemical constituents, traditional use and modern pharmacological studies of T. chinense reported in the literature in the first paragraph of Introduction section (lines 33-39).
- The authors could explain why did not test antioxidant, anti-inflamnatory and citotoxic effects of supstance 1.
Response: Thank you very much for your carefully review and kindly suggestion, and we were not scanned for these activities of compound 1 in that the weight of compound 1 (just 1.8 mg) was too small to do related experiments after structural identification. We have specified the meaning of nd and nc below Table 3-5.
Round 2
Reviewer 1 Report
The author addressed the comments and they included the missing results; and recommend for publication
Author Response
Thank you.
Reviewer 2 Report
I believe that the manuscript was improved after carrying out the recommended suggestions. However, I believe that the Structure-activity relationship discussion of all biological results should be improved
Author Response
Response: We appreciate this comment very much, which have contributed to an improvement of this paper. We've talked more about the structure-activity relationship discussion of all biological results. The comment resulted changes in Conclusions (line 302-329) in the revision.